

**Evaluating the streamflow simulation capability of PERSIANN-CDR daily**
**rainfall products in two river basins on the Tibetan Plateau**
Xiaomang Liu[1,2], Tiantian Yang[2], Koulin Hsu[2], Changming Liu[1] and Soroosh
Sorooshian[2]
1 Key Laboratory of Water Cycle & Related Land Surface Process, Institute of Geographic
Sciences and Natural Resources Research, Chinese Academy of Sciences, 100101 Beijing, China
2 Department of Civil and Environmental Engineering, University of California, Irvine, California,
USA
*Corresponding author: Tiantian Yang, Email: tiantiay@uci.edu



Abstract:
On the Tibetan Plateau, the limited ground-based rainfall information owing to a harsh
environment has brought great challenges to hydrological studies. Satellite-based rainfall products,
which allow a better coverage than both radar network and rain gauges on the Tibetan Plateau, can
be suitable observation alternatives for investigating the hydrological processes and climate
change. In this study, a newly developed daily satellite-based precipitation product, termed
Precipitation Estimation from Remotely Sensed Information Using Artificial Neural Networks–
Climate Data Record (PERSIANN-CDR), is used as input of a hydrologic model to simulate
streamflow in the upper Yellow and Yangtze River Basin on the Tibetan Plateau. The results show
that the simulated streamflow using PERSIANN-CDR precipitation is closer to observation than
that using limited gauge-based precipitation interpolation in the upper Yangtze River Basin. The
simulated streamflow using gauge-based precipitation are higher than the streamflow observation
during the wet season. In the upper Yellow River Basin, PERSIANN-CDR precipitation and
gauge-based precipitation have similar good performance in simulating streamflow. The
evaluation of streamflow simulation capability in this study partly indicates that PERSIANN-CDR
rainfall product has good potentials to be a reliable dataset and an alternative information source
besides the sparse gauge network for conducting long term hydrological and climate studies on the
Tibetan Plateau.
Key Words: PERSIANN-CDR daily rainfall product; Streamflow simulation; Tibetan Plateau
**1. Introduction**

Precipitation is one of the essential meteorological inputs of hydrologic model and the key

driving force for hydrologic cycle. Errors in precipitation estimation can bring significant
uncertainties in streamflow simulation and prediction (Sorooshian et al., 2011). Three methods are





generally used to measure precipitation: traditional gauge observations, meteorological radar
observations and satellite observations (Ashouri et al., 2015). In many remote regions and
mountainous area, rain gauges and meteorological radar networks are either sparse or non-existent.
Thus, satellite-based precipitation is of great importance in such regions. For instance, there is a
great potential of using satellite-based precipitation estimate on the Tibetan Plateau known as the
"roof of the world" with an average elevation of over 4000m (Yao et al., 2012). Owing to a harsh
environment, the existing meteorological stations managed by the Chinese Meteorological
Administration only form an extremely sparse network, which create great challenges for water
resources management and operation. For example, on average, there is only 0.3 and 1 station per
grid of 1°×1° in the upper Yangtze and upper Yellow river basins, respectively. Moreover, the
spatial distribution of the meteorological stations is highly uneven and most stations are located
around the river channel with relatively low elevation [Figure 1]. Therefore, streamflow simulation
using the limited gauge-based rainfall information might not be reliable due to the input
uncertainties with such a poor spatial resolution. Satellite-based rainfall products have the
advantage of good spatial coverage, which could allow an accurate streamflow simulation on the
Tibetan Plateau.
According to Kidd and Levizzani (2011), during the last decade satellite-based precipitation
estimates have reached a good level of maturity. Currently, there are many satellite rainfall
products are available and have been extensively used globally (e.g., Sorooshian et al., 2000;
Huffman et al., 2001; Adler et al., 2003; Xie et al., 2003; Joyce et al., 2004; Turk and Miller, 2005).
Recently, a new satellite-based precipitation product is released by National Climatic Data Center
(NCDC), which is termed Precipitation Estimation from Remotely Sensed Information Using
Artificial Neural Networks–Climate Data Record (PERSIANN-CDR) (Ashouri et al., 2015).



PERSIANN-CDR is a multi-satellite, high-resolution and post-time rainfall product that provides
daily precipitation estimates at 0.25˚ spatial resolution from 1 January 1983 to the present.
According to Ashouri et al., (2015), PERSIANN-CDR rainfall product uses the archive of Gridded
Satellite (GridSat-B1) Infrared Radiation (IR) data (Knapp, 2008) as the input to the Artificial
Neural Network algorithm. The retrieval algorithm uses IR satellite data from global
geosynchronous satellites as the primary source of precipitation information. To meet the
calibration requirement of PERSIANN, the model is pre-trained using the National Centers for
Environmental Prediction (NCEP) stage IV hourly precipitation data. Then, the parameters of the
model are kept fixed and the model is run for the full historical record of GridSat-B1 IR data. To
reduce the biases in the estimated precipitation, while preserving the temporal and spatial patterns
in high resolution, the resulting estimates are then adjusted using the Global Precipitation
Climatology Project (GPCP) monthly 2.5˚ precipitation products. The performance of
PERSIANN-CDR rainfall product has been tested and reported in different regions (Ashouri et al.
2015; Casse et al. 2015; Miao et al., 2015). Ashouri et al. (2015) found that PERSIANN-CDR
precipitation is performing reasonably well when compared with radar and ground-based
observations in the 1986 Sydney flood event of Australia and the 2005 Hurricane Katrina of United
States. Miao et al. (2015) shows that PERSIANN-CDR rainfall product is able to capture the
spatial and temporal characteristics of extreme precipitation events at daily scale in the eastern
China monsoon region when compared with ground-based precipitation dataset. Casse et al. (2015)
employed PERSIANN-CDR to simulate the flood events in Niamey River and compared with
multiple datasets. It was found out by Casse et al. (2015) that the simulated streamflow with
PERSIANN-CDR has high correlation as compared to streamflow observation. Miao et al. (2015)
also pointed out that the correlation between the PERSIANN-CDR precipitation and ground-based





precipitation is not strong on the Tibetan Plateau and speculated that the sparse ground-based
gauge stations may result in uncertainties of the use of ground-based precipitation estimates as
reference on the Tibetan Plateau. Building on Miao et al. (2015), in this study, PERSIANN-CDR
is further applied to a conceptual hydrological model to simulate streamflow of two river basins
on the Tibetan Plateau. The simulated streamflow with PERSIANN-CDR is compared with that
forced by limited gauge information, as well as the streamflow observations at the outlets of these
two basins.
Many studies have been carried out to evaluate the suitability of a number of satellite-based
precipitation estimate products in forcing hydrologic models and simulating streamflow for
various regions around the world (e.g., Yilmaz et al., 2005; Artan et al., 2007; Su et al., 2011; Bitew
et al., 2012; Yong et al., 2012). However, there are few evaluation works focusing on hydrological
modeling driven by satellite rainfall products on the Tibetan Plateau. Among limited number of
studies, Tong et al. (2014) evaluated the streamflow simulation capability of four satellite products
(TRMM-3B42-V7, TRMM-3B42RT-V7, PERSIANN and CMORPH) using the Variable
Infiltration Capacity (VIC) hydrologic model in two sub-basins over the Tibetan Plateau and
concluded that the TRMM-3B42-V7 and CMORPH datasets have relative better performance than
others. One of the limitations is that the data of many satellite precipitation products, such as
TRMM-3B42RT-V7 and CMORPH only starts from 2000 to the present, which is rather short. In
this study, there is no such limitation because PERSIANN-CDR daily rainfall product includes
more than 33 years of data and the length of data grows every year. In Tong et al, (2014), the rain-
gauge is set to be reference to compare different satellite-based rainfall products. However, given
the facts that (1) density of rain-gauges on Tibetan Plateau is rather low as compared to other
regions in China, (2) distribution of gauges are uneven according to the investigation from Miao



et al, (2015), and (3) rain-gauges are located in low elevation river channels (Figure 1), authors
have the similar concern as Miao et al, (2015) that the use of sparse rain-gauge network as
reference to compare satellite products is questionable. Therefore, in this study, both PERSIANN-
CDR daily rainfall product and ground-based rainfall information are used as the inputs of a
hydrologic model for streamflow simulation on two major river basins, the upper Yangtze River
Basin and the upper Yellow River Basin, on the Tibetan Plateau. Then, the simulation results are
compared with observed streamflow, which is believed to be a more reliable reference than the
limited rainfall observation to judge the qualities of satellite rainfall products on the Tibetan
Plateau. The uncertainty sources are also discussed with regard to the parameterization of
hydrological model and the length of data used for calibration.

## 127    2. Study region, data and hydrological modeling

### 128    2.1 Study region and data

Two river basins on the northern Tibetan Plateau, namely, the upper Yangtze River (UYZR)
and upper Yellow River (UYLR) basins are selected, which have a long daily streamflow record
from 1983 to 2012. As shown with red squares in Figure 1, two hydrological stations, Tangnaihai
and Zhimenda, are the outlet stations of the UYZR and UYLR, which have total drainage areas of
121,972 and 137,704 km$^2$, respectively. Elevation in the region varies from 3450 to 6621 m.
According to Yao et al. (2012), the climate system of the two selected regions have distinct summer
Indian monsoon and East Asian monsoon characteristics during summer. Figure 1 shows the
distribution of meteorological and hydrological stations in the two basins. The green triangles show
the locations of rain-gauges, which are rather unevenly distributed and sparse as compared to the
gauge distribution of in other regions of China (Miao et al., 2015).





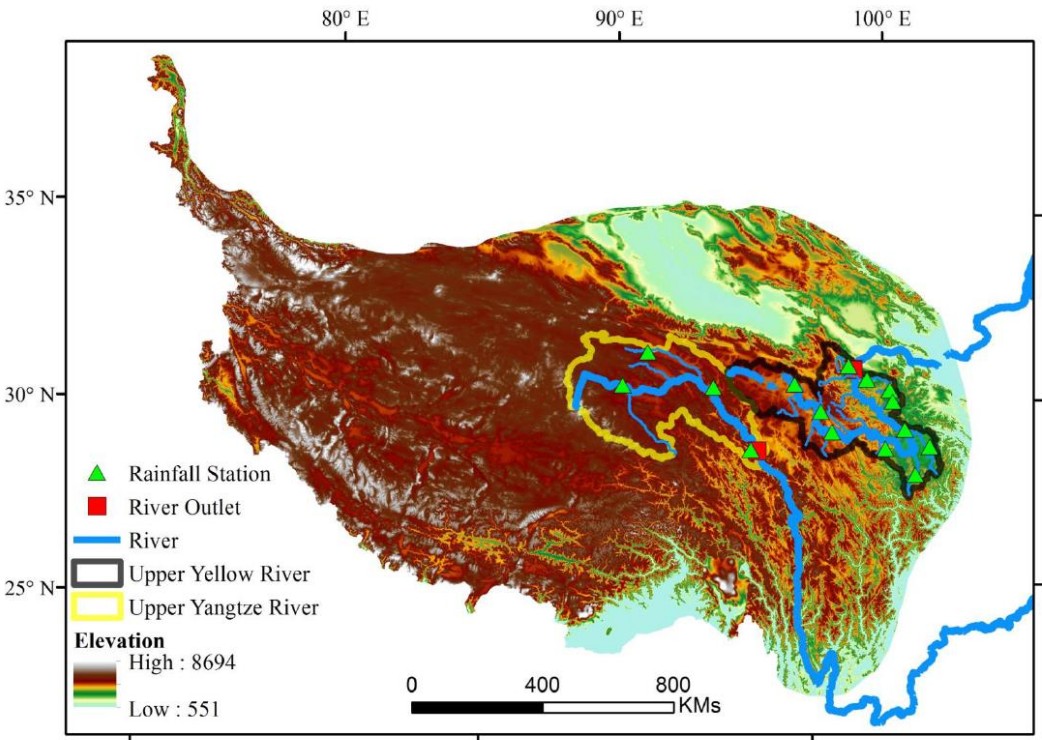

Figure 1. The selected river basins (the upper Yellow River and Yangtze River Basin) on the

Tibetan Plateau and location of rainfall stations and river outlets.

The observed daily streamflow data from 1983 to 2012 at the outlets of the two basins is provided

by the Ministry of Water Resources of China. The runoff is calculated by dividing streamflow by

corresponding basin area. The daily gauge meteorological data in the two basins from 1983 to

2012 is obtained from the China Meteorological Administration (http://cdc.cma.gov.cn). There are

4 and 11 meteorological stations in the UYZR and UYLR respectively, which means that on

average there is only 0.3 and 1 station per grid of $1° \times 1°$ in the two basins, respectively. The

PERSIANN-CDR rainfall dataset is available at the NOAA NCDC website

(ftp://data.ncdc.noaa.gov/cdr/persiann/files/), as well as the Center for Hydrometeorology and



Remote Sensing (CHRS) at the University of California, Irvine. In order to compare PERSIANN-
CDR with gauge observation, the gauge precipitation is interpolated into 0.25˚×0.25˚ grids with
the inverse distance weighting interpolation method, which has been demonstrated as being
efficient in precipitation interpolation applications (e.g., Nalder and Wein, 1998; Garcia et al., 2008;
Ly et al., 2011). The daily gauge-based precipitation and PERSIANN-CDR precipitation for basin
average are compared by the cumulative distribution functions (CDFs) of daily precipitation value
(e.g., Sheffield et al., 2014; Zhang and Tang, 2015), wherein the two-parameter Gamma
distribution function (Thom, 1958) is used to fit the data.
**2.2 Hydrological modeling**
The hydrologic model used in this study is the Hydro-Informatic Modeling System (HIMS)
rainfall-runoff model (Liu et al., 2006, 2008, 2010a, 2010b), which is one of the operational
hydrological models in practice by the Tibet Government in China. The HIMS model is a grid-
based hydrologic model, which is able to simulate the dominant hydrological processes such as
actual evapotranspiration, infiltration, runoff, groundwater recharge and channel routing. In HIMS
model, a catchment is divided into grids, and grids are linked throughout the stream network based
on topological relationships of channel network and properties of soil, vegetation and land use. In
each grid, actual evaporation is calculated by a formulation between soil water content and
potential evapotranspiration. Potential evapotranspiration $ET_0$ (Hargreaves and Samani, 1985) and
actual evaporation $ET_a$ are described as follows:

$$ET_0 = 0.00023 \cdot RA \cdot (T + 17.8) \cdot (T_{max} - T_{min})^{0.50} \tag{1}$$

$$ET_a(t) = ET_0(t) \cdot (1 - (1 - \frac{SMS_t}{SMSC})^C) \tag{2}$$

where $RA$ is extraterrestrial radiation (MJ m$^{-2}$ d$^{-1}$); $T$, $T_{max}$ and $T_{min}$ are daily average, maximum
and minimum temperatures, respectively (°C); $L$ is latent heat of vaporization (MJ kg$^{-1}$); $SMS$ and





*SMSC* are soil moisture storage and the maximum soil storage capacity (mm), respectively; and *C*
is the evapotranspiration coefficient in need of calibration.

Infiltration is modeled using an empirical relationship, which has been confirmed through

analysis of data measured in a number of experimental watersheds and various physical geographic
factors in China (Liu et al., 2006):
$$f_t = R \cdot P_t^r \tag{3}$$
where $f_t$ is infiltration (mm) and $P_t$ is precipitation (mm). *R* and *r* are parameters. Surface runoff
$RS_t$ (mm) is calculated by:
$$RS_t = P_t - f_t = P_t - R \cdot P_t^r \tag{4}$$

According to the saturation excess mechanism and spatial variability of watershed

characteristics, interflow and groundwater recharge are estimated as linear functions of soil
wetness (soil moisture amount divided by soil moisture capacity). Baseflow is simulated based on
the linear reservoir assumption, in which the relationship between groundwater storage and
outflow is linear. Interflow *RI* (mm), groundwater recharge *REC* (mm), baseflow *RG* (mm), and
total runoff *TR* (mm) are determined by:
$$RI_t = L_a \times (SMS_t / SMSC) \times f_t \tag{5}$$
$$REC_t = R_C \times (SMS_t / SMSC) \times (f_t - RI_t) \tag{6}$$
$$RG_t = K_b \times (GW_t + REC_t) \tag{7}$$
$$TR_t = RS_t + RI_t + RG_t \tag{8}$$
where $L_a$, $R_c$ and $K_b$ are efficiencies for interflow, groundwater recharge and baseflow, respectively;
*SMSC* is the maximum value of soil moisture storage capacity (mm); *SMS* is actual soil moisture
storage (mm); and *GW* is groundwater storage (mm). $L_a$, $R_c$, $K_b$ and *SMSC* are parameters in need



of calibration. The degree-day snowmelt algorithm (Hock, 2003) assuming an empirical
relationship between air temperature and snowmelt rate is used to simulate snowmelt runoff. The
air temperature within each grid is adjusted by a commonly used temperature lapse rate
(0.65°C/100m). The degree-day factor of snowmelt is set to 4.1 mm°C$^{-1}$ day$^{-1}$ in the two basins
based on the investigation of Zhang et al. (2006). Surface runoff and baseflow for each grid are
routed to the basin outlet through a channel network. The Muskingum method (Franchini and
Lamberti, 1994) is used for flow channel routing. The detail descriptions of HIMS model refer to
Liu et al. (2008) and Jiang et al. (2015).
The HIMS model is set up at 0.25°×0.25° spatial resolution grids in the two river basins.
There are nine parameters requiring calibration in the HIMS model (Table 1). The Shuffle Complex
Evolution method (SCE-UA) is used for calibrating the model parameters (Duan et al., 1992). The
optimization objective is to maximize the Nash-Sutcliffe efficiency (*NSE*) (Nash and Sutcliffe,
1970) between the simulated and measured daily streamflow. There are two stopping criteria for
calibrating the parameters. The first one is the evolution of all simplexes have converged to a
limited parameter space, which is the default convergence criterion of SCE-UA. Another stopping
criterion is the maximum number of function evaluation set by users is met. In our study, the
settings for SCE-UA are: maximum number of function evaluation equals to $5\times10^8$; numbers of
complexes equals to 2; and the percentage change allowed to define convergence is set to $1\times10^{-6}$.
The calibration period is from 1983 to 1997 and the verification period is from 1998 to 2012. The
performance of the streamflow simulation is evaluated by comparing simulated and observed
streamflow through two statistics: *NSE* and relative bias (*Rb*) between simulated and observed
streamflow:



$$NSE = 1 - \frac{\sum_{i=1}^{N}(Q_{obs,i} - Q_{sim,i})^2}{\sum_{i=1}^{N}(Q_{obs,i} - \overline{Q_{obs}})^2}$$
(9)

$$Rb = \frac{\sum_{i=1}^{N}(Q_{sim,i} - Q_{obs,i})}{\sum_{i=1}^{N}Q_{obs,i}}$$
(10)

where $Q_{sim}$ and $Q_{obs}$ are the simulated and observed streamflow, respectively; $\overline{Q_{obs}}$ is the mean of
the observed streamflow; and $N$ is the total number of days in the calibration period.

Table 1. Description of HIMS model parameters and allowable ranges.

| Parameter | Description | Allowable range |
|---|---|---|
| $SMSC$ | The maximum soil storage capacity (mm) | 50-1000 |
| $R$ | The infiltration coefficient | 0.1-2 |
| $r$ | The infiltration coefficient | 0.1-1 |
| $L_a$ | The interflow coefficient | 0.1-2 |
| $R_C$ | The groundwater recharge coefficient | 0.01-2 |
| $C$ | The evapotranspiration coefficient | 0.001-10 |
| $K_b$ | The baseflow coefficient | 0.001-1 |
| $C_1$ | The Muskingum coefficient | 0.001-1 |
| $C_2$ | The Muskingum coefficient | 0.001-1 |


## 3. Results

### 3.1 Hydrometrological characteristics of the two basins

Figure 2 and Table 2 show the average monthly amounts of precipitation and runoff in the
UYZR and UYLR from 1983 to 2012. These two river basins have distinct dry and wet seasons,
which are from Sep. to Feb., and Mar. to Oct., respectively. According to Table 2, precipitation
between May and October (wet season) accounts for 92.5% and 90.1% of the annual total
precipitation for the UYZR and UYLR, respectively. . Similar to the temporal distribution of





precipitation, runoff during May to October accounts for 87.6% and 78.4% of annual runoff in the
two basins, respectively. Given the seasonal concurrence of precipitation and runoff, thus,
precipitation in wet season plays a dominant role in annual runoff generation in these two river
basins. The runoff coefficients are 0.22 and 0.26 in the UYZR based on gauge-based precipitation
and PERSIANN-CDR precipitation, respectively. In the UYLR, the runoff coefficient is 0.29 based
on both gauge-based and PERSIANN-CDR precipitation.

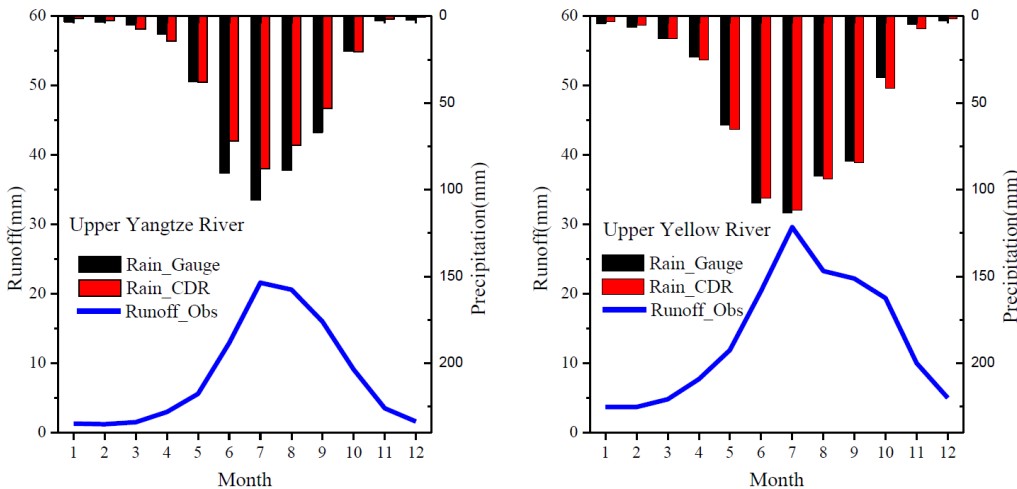


Figure 2. The monthly average runoff observed at the river outlet of the upper Yangtze River and
Yellow River Basin, and the precipitation data retrieved from PERSIANN-CDR and ground-based
observation.

**3.2 Comparison between gauge-based precipitation and PERSIANN-CDR precipitation**
Figure 3 shows the comparison of CDFs for basin-averaged daily gauge-based precipitation
and PERSIANN-CDR precipitation in the UYZR and UYLR from 1983 to 2012. At a given
probability, daily precipitation of PERSIANN-CDR product is generally smaller than gauge-based
precipitation in the UYZR. In the UYLR, the CDFs of PERSIANN-CDR precipitation and gauge-





based precipitation show overall better agreement than that in the UYZR. Table 2 shows the
average monthly amounts of gauge-based precipitation and PERSIANN-CDR precipitation.
Gauge-based precipitation is larger than PERSIANN-CDR precipitation in wet season (Jun., Jul.,
Aug. and Sep.) in the UYZR, while gauge-based precipitation and PERSIANN-CDR precipitation
have similar values in the UYLR. In the UYZR, the average annual precipitation is 436.4 mm from
gauge-based data and 374.3 mm from PERSIANN-CDR product, and the former is 16.6% larger
than the latter. In the UYLR, average annual amounts of gauge-based precipitation and
PERSIANN-CDR precipitation are similar, which are 550.2 and 556.6 mm, respectively (Table 2).

Table 2. Average monthly precipitation and runoff in the upper Yangtze and Yellow River basins

| Period | Upper Yangtze River | | | Upper Yellow River | | |
|---|---|---|---|---|---|---|
| | Rain_ Gauge | Rain_ CDR | Runoff_ OBS | Rain_ Gauge | Rain_ CDR | Runoff_ OBS |
| Jan. | 3.3 | 1.4 | 1.3 | 4.4 | 3.2 | 3.7 |
| Feb. | 3.4 | 2.5 | 1.2 | 6.5 | 5.2 | 3.7 |
| Mar. | 5.0 | 7.5 | 1.5 | 12.9 | 13.1 | 4.8 |
| Apr. | 10.2 | 14.6 | 3.0 | 23.7 | 25.0 | 7.7 |
| May | 37.9 | 38.2 | 5.6 | 62.9 | 65.3 | 11.9 |
| Jun. | 90.4 | 72.0 | 12.9 | 107.6 | 104.6 | 20.4 |
| Jul. | 105.8 | 87.8 | 21.6 | 113.5 | 111.8 | 29.6 |
| Aug. | 88.6 | 74.5 | 20.6 | 92.0 | 94.0 | 23.3 |
| Sep. | 66.9 | 53.2 | 16.0 | 83.4 | 84.4 | 22.2 |
| Oct. | 20.2 | 20.5 | 9.1 | 35.3 | 41.4 | 19.4 |
| Nov. | 2.5 | 1.7 | 3.5 | 5.0 | 7.3 | 10.0 |
| Dec. | 2.3 | 0.5 | 1.6 | 3.0 | 1.5 | 5.0 |
| May to Oct. | 409.7 | 346.1 | 85.8 | 494.6 | 501.4 | 126.8 |
| Annual | 436.4 | 374.3 | 98.0 | 550.2 | 556.6 | 161.8 |
| Ratio | 93.9 | 92.5 | 87.6 | 89.9 | 90.1 | 78.4 |

*Note: Rain_Gauge and Rain_CDR indicate gauge-based precipitation and PERSIANN-CDR*
*precipitation (mm), respectively. Runoff_OBS indicates observed runoff (mm). Ratio means the*
*percentage of precipitation and streamflow during May to November to annual values.*




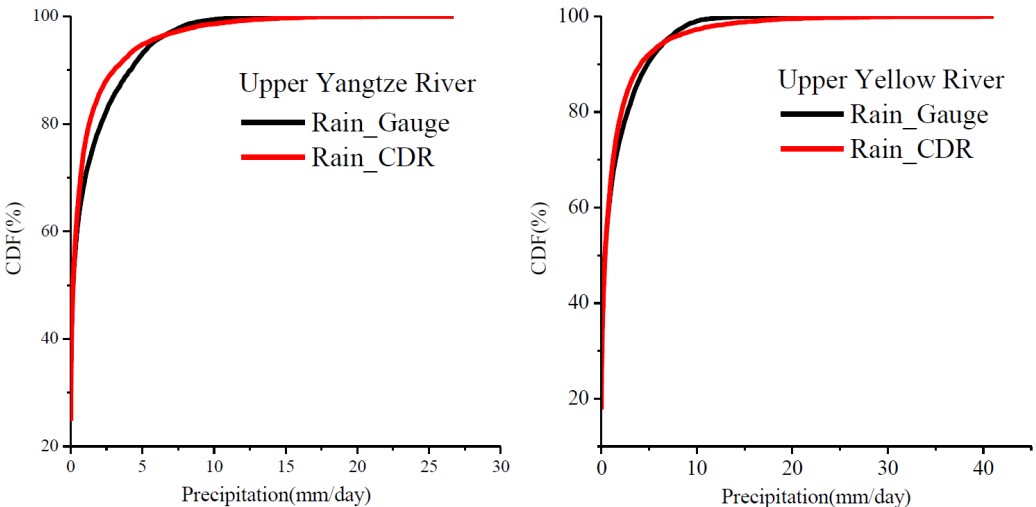


Figure 3. The calculated CDF of rainfall from PERSIANN-CDR and ground-based observation in

the upper Yangtze River Basin and upper Yellow River Basin.



**3.3  Streamflow Simulation in the two basins**

Due to the fact that sparse gauge-network and its interpolation cannot perfectly describe the

spatial and temporal rainfall characteristics at river basin scale, the alternative is to evaluate
streamflow simulated instead of over-confidently using sparse gauge-network as reference. In this
section, the streamflow simulated by both gauge-based precipitation and PERSIANN-CDR
precipitation are derived from HIMS model, and compared with observed streamflow at the outlet
in the UYZR and UYLR. The HIMS model is separately calibrated by maximizing the *NSE*
between observed streamflow and simulated streamflow driven by gauge-based precipitation and
PERSIANN-CDR precipitation from 1983 to 1997. Table 3 shows the calibrated parameter values
of the HIMS model for the two basins. Figure 4 shows daily observed streamflow and simulated
streamflow driven by gauge-based precipitation and PERSIANN-CDR precipitation for the two





basins from 1983 to 2012. In the UYLR (Figure 4 (a) and (b)), the *NSE* values between daily
observed streamflow and simulated streamflow are 0.82 and 0.80 in the calibration period driven
by gauge-based precipitation and PERSIANN-CDR precipitation, respectively. In the verification
period, the *NSE* values are 0.81 and 0.78 for the two types of data, respectively. The high *NSE*
value in both calibration and verification periods suggest that gauge-based precipitation and
PERSIANN-CDR precipitation have similar performances as the drivers of streamflow simulation
in the UYLR. In the UYZR (Figure 4 (c) and (d)), the *NSE* values are 0.63 and 0.77 in the
calibration period driven by gauge-based precipitation and PERSIANN-CDR precipitation
respectively, while they are 0.60 and 0.73 in the verification period, respectively. In both
calibration and verification period, the *NSE* values from PERSIANN-CDR precipitation are
greater than that from gauge-based precipitation, which indicates that using PERSIANN-CDR
precipitation as input to HIMS model is able to generate more accurate streamflow than using the
interpolated precipitation from a sparse gauge-network in the UYZR.







Figure 4. The comparison between the simulated daily streamflow (red) with PERSIANN-CDR





and ground-based precipitation and the observed data (black) at the outlets of the upper Yangtze
River Basin (a and b) and upper Yellow River Basin (c and d).

Table 3. Calibrated parameter values in the HIMS model for the upper Yangtze and Yellow River
basins.

| Basin | input | $SMSC$ | $R$ | $r$ | $L_a$ | $R_c$ | $C$ | $K_b$ | $C_1$ | $C_2$ |
|---|---|---|---|---|---|---|---|---|---|---|
| Yangtze | Gauge_based | 302.46 | 1.47 | 0.78 | 0.74 | 0.05 | 0.67 | 0.15 | 0.18 | 0.81 |
| | PERSIANN-CDR | 343.80 | 1.71 | 0.89 | 0.87 | 0.07 | 0.56 | 0.18 | 0.17 | 0.82 |
| Yellow | Gauge_based | 334.82 | 2.08 | 0.77 | 1.00 | 0.03 | 0.44 | 0.14 | 0.14 | 0.86 |
| | PERSIANN-CDR | 342.08 | 2.01 | 0.73 | 0.98 | 0.05 | 0.45 | 0.14 | 0.12 | 0.88 |


Figure 5 and Table 4 compare the simulated and observed average monthly streamflow for

the two basins. In the UYZR, the relative bias between observed streamflow and simulated
streamflow driven by gauge-based precipitation is 10.3% in wet season, which suggests a
considerable overestimate of streamflow. Comparably, the relative bias between observed
streamflow and simulated streamflow driven by PERSIANN-CDR precipitation is 0.5% in wet
season. As compared with the wet season streamflow simulation results with gauge-based
precipitation, the simulated streamflow driven by PERSIANN-CDR precipitation is closer to the
observed streamflow. In dry season, streamflow simulations driven by gauge-based precipitation
and PERSIANN-CDR precipitation both underestimate streamflow with relative bias of -22.1 and
-28.0% in the UYZR, respectively. In the UYLR, both the two precipitation products slightly
overestimate the streamflow in wet season with relative bias of 2.6 and 2.9%, respectively. Similar
to the results in the UYZR, streamflow simulations driven by gauge-based precipitation and
PERSIANN-CDR precipitation have similar good performances in wet season in the UYLR.



However, both the two precipitation products tend to produce smaller streamflow in dry season
with relative bias of -33.1% and -27.6%, respectively. One of the reasons that both PERSIANN-
CDR and gauge-based precipitation generate smaller streamflow in dry season is the lack of
complex method or proper algorithm in the HIMS model to handle frozen soil. In dry season, when
the amounts of precipitation and streamflow are small, streamflow melted from frozen soil can
account for a significant proportion of total streamflow. In other words, the frozen soil melt could
significantly influence the streamflow simulation results. The relative high bias of observed
streamflow and simulated streamflow from both gauge-based and satellite-based precipitation
could be due to the lack of proper modeling component in the HIMS hydrologic model that
quantifies the frozen soil melting effects in dry season. However, the bias between simulated and
observed streamflow is much smaller in wet season, when precipitation and streamflow are
relatively large and streamflow melted from frozen soil accounts for a limited proportion in total
streamflow.
In summary, the streamflow simulated by PERSIANN-CDR precipitation has a good
agreement with the observed streamflow in the UYZR and UYLR. The good agreement between
observed streamflow and PERSIANN-CDR simulated streamflow reveals a strong streamflow
simulation capability of PERSIAN-CDR product, which also gives community certain confidence
in using PERSIANN-CDR product to study hydrological cycle and climate change on the Tibetan
Plateau.



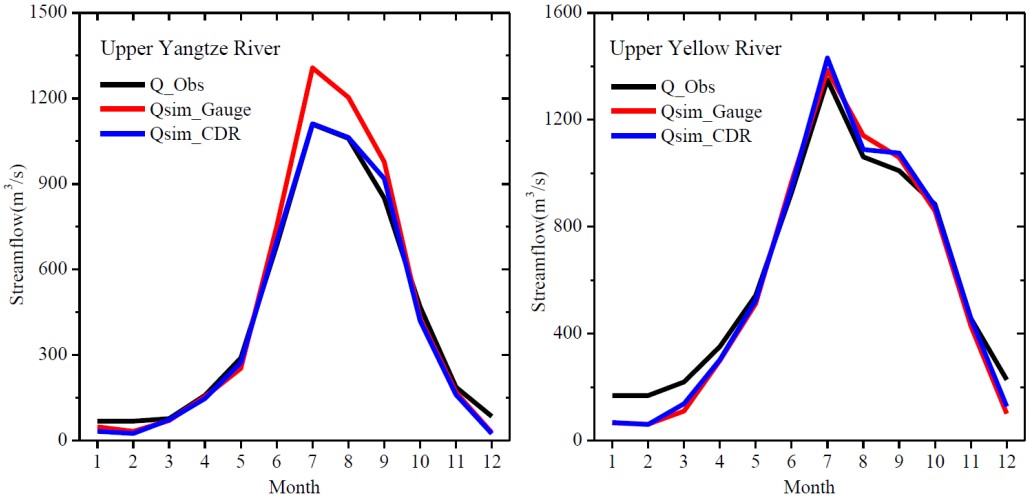


Figure 5. The comparison between the observed streamflow (black) and the simulated streamflow

using PERSIANN-CDR (blue) and ground-based precipitation data (red) in the upper Yangtze

River Basin and upper Yellow River Basin.

Table 4. The performances of streamflow simulations driven by gauge-based precipitation

and PERSIANN-CDR precipitation in the two basins

| Period | Upper Yangtze River | | | | | Upper Yellow River | | | | |
|---|---|---|---|---|---|---|---|---|---|---|
| | Q_obs | Qsim_ gauge | Qsim_ CDR | Rb_ gauge | Rb _ CDR | Q_obs | Qsim_ gauge | Qsim_ CDR | Rb _ gauge | Rb _ CDR |
| Jan. | 68.1 | 48.4 | 32.8 | -28.9 | -51.8 | 168.9 | 65.7 | 68.0 | -61.1 | -59.8 |
| Feb. | 68.3 | 32.7 | 24.9 | -52.1 | -63.5 | 168.3 | 61.6 | 60.5 | -63.4 | -64.1 |
| Mar. | 76.9 | 70.2 | 72.4 | -8.7 | -5.8 | 219.7 | 110.5 | 138.0 | -49.7 | -37.2 |
| Apr. | 158.6 | 153.2 | 147.5 | -3.4 | -7.0 | 352.0 | 299.0 | 302.5 | -15.1 | -14.0 |
| May | 289.2 | 253.5 | 273.4 | -12.3 | -5.5 | 543.6 | 512.9 | 524.9 | -5.7 | -3.4 |
| Jun. | 683.9 | 750.5 | 698.4 | 9.7 | 2.1 | 928.5 | 968.6 | 946.6 | 4.3 | 1.9 |
| Jul. | 1108.9 | 1306.9 | 1111.4 | 17.9 | 0.2 | 1350.1 | 1386.6 | 1431.3 | 2.7 | 6.0 |
| Aug. | 1059.7 | 1204.0 | 1063.2 | 13.6 | 0.3 | 1061.1 | 1141.4 | 1088.5 | 7.6 | 2.6 |
| Sep. | 850.7 | 977.4 | 918.9 | 14.9 | 8.0 | 1009.6 | 1059.7 | 1075.7 | 5.0 | 6.5 |
| Oct. | 469.4 | 428.1 | 420.1 | -8.8 | -10.5 | 883.7 | 859.1 | 876.5 | -2.8 | -0.8 |
| Nov. | 187.6 | 169.0 | 161.1 | -9.9 | -14.1 | 457.3 | 429.1 | 456.6 | -6.2 | -0.2 |
| Dec. | 84.5 | 28.2 | 24.5 | -66.7 | -71.0 | 227.0 | 100.7 | 127.5 | -55.7 | -43.9 |
| May-Oct. | 743.4 | 819.6 | 746.9 | 10.3 | 0.5 | 962.7 | 987.7 | 990.4 | 2.6 | 2.9 |
| Nov.-Apr. | 107.2 | 83.6 | 77.2 | -22.1 | -28.0 | 265.6 | 177.6 | 192.3 | -33.1 | -27.6 |
| Annual | 427.9 | 454.6 | 414.8 | 6.2 | -3.1 | 617.0 | 586.0 | 594.6 | -5.0 | -3.6 |





*Note: Q_obs indicates observed runoff (mm). Qsim_gauge and Qsim_CDR indicate*
*streamflow simulations (mm) driven by the gauge-based precipitation and PERSIANN-CDR*
*precipitation, respectively. Rb_gauge and Rb_CDR indicate relative bias between observed*
*streamflow and simulated streamflow driven by the gauge-based precipitation and*
*PERSIANN-CDR precipitation, respectively.*

## 4. Discussions

### 4.1 Parameter uncertainties of hydrological modeling

In this study, model parameters are separately calibrated in terms of highest *NSE* between
observed streamflow and simulated streamflow driven by gauge-based precipitation and
PERSIANN-CDR precipitation. Therefore, these parameter values are highly dependent on the
precipitation inputs. When the precipitation input changes, the parameter values may change
accordingly in order to match the streamflow. Table 3 shows the values of calibrated parameters
separately driven by gauge-based precipitation and PERSIANN-CDR precipitation in the two
basins. Parameter sensitivity study of the HIMS model indicates that the HIMS model is most
sensitive to parameters of the maximum soil storage capacity (*SMSC*) and the infiltration
coefficients (*R* and *r*) (Jiang et al., 2015). In the UYLR, the parameters calibrated by the inputs of
gauge-based precipitation and PERSIANN-CDR precipitation generally have similar values.
However, in the UYZR, *SMSC*, *R* and *r* values calibrated from gauge-based precipitation are
302.46, 1.47 and 0.78 respectively, while *SMSC*, *R* and *r* values calibrated from PERSIANN-CDR
precipitation are 343.80, 1.71 and 0.89 respectively. By separately calibrating the parameters in
HIMS model, gauge-based precipitation and PERSIANN-CDR result in different optimal
parameter values. . Thus, the streamflow simulation bias using gauge-based precipitation and
PERSIANN-CDR are the joint results of parameter differences and model input bias.
Correspondingly, soil moisture and evapotranspiration estimation could be different using various





precipitation forcings and calibrated parameters. However, the main purpose of this study is

evaluating the streamflow simulation capability of satellite-based precipitation and gauge-based

precipitation as inputs to a hydrologic model over the Tibetan Plateau. Therefore, in spite of the

influence of cancellation between parameter differences and precipitation bias on streamflow

simulation, it does not harm the conclusion that PERSIANN-CDR precipitation is able to produce

a reasonably good streamflow in the two river basins on the Tibetan Plateau.

In previous study, Tong et al. (2014) evaluated the streamflow simulation capabilities of four

satellite-based precipitation products (TRMM-3B42-V7, TRMM-3B42RT-V7, PERSIANN and

CMORPH) using the VIC hydrologic model in the UYZR and UYLR from 2006 to 2012. Different

from the PERSIANN product that Tong et al. (2014) used, PERSIANN-CDR is a different product

that provides over 33 years of daily and high resolution precipitation with GPCP monthly

information incorporated. In addition, the parameters in the VIC hydrologic model are calibrated

by the input of interpolated gauge-based precipitation. The calibrated parameter values are then

kept fixed when the VIC model are rerun by inputs of satellite-based precipitation datasets to

evaluate the streamflow simulation capabilities of satellite-based precipitation datasets. Rerunning

the hydrologic model with the fixed parameters calibrated by gauge-based precipitation partly

indicates that the authors consider the sparse gauge observations a more reliable dataset than

satellite-based precipitation datasets. However, this is a questionable assumption. As we mentioned

in the introduction, not only because the location of rain-gauges is conditioned (relatively low

elevations), but also the sparse distribution of rainfall stations over the Tibetan Plateau could bring

large errors and uncertainties in regional rainfall measurement. We rather cautiously believe that

gauge-based precipitation could not be reliable, especially in the UYZR where there is only one

station per 34426 $km^2$ (nearly 1°×3° spatial resolution). Therefore, separately calibrating





hydrologic model by the inputs of different precipitation datasets instead of using identical
parameters will contribute to fairer comparisons when evaluating streamflow simulation
capabilities of different precipitation datasets, though other hydrological variables such as soil
moisture and evapotranspiration could be incorrectly estimated by different precipitation inputs
and calibrated parameters.
**4.2  The influences of precipitation record length on streamflow simulation capability**
Besides of the uncertainties due to hydrological model calibration, another factor that
influences the accuracy of streamflow simulation is the length of precipitation records used for
calibration. As mentioned before, one of the advantages of PERSIANN-CDR product is the
provision of more than 33 years of continuous sequences of precipitation data, which can allow
more extensive streamflow simulation in the Tibetan Plateau. In this study, comparison
experiments (Figure 6) were designed to test the influences of precipitation record length on the
accuracy of streamflow simulation. In the experiments, we investigate the accuracy of streamflow
simulation during 2008 to 2012 with two different calibration scenarios. In the first scenario, the
calibration period is from 2003 to 2007 for both the UYZR (Figure 6a) and the UYLR (Figure 6b).
In the second scenario (Figure 6c and 6d), 15 years of data from 1983 to 1997 is used for calibration,
which is longer than that in the first scenario. As it is shown in Figure 6 (a and b), in the first
scenario the *NSE* values between daily observed and simulated streamflow are 0.75 and 0.66
during the verification period (from 2008 to 2012) for the UYZR and UYLR, respectively.
Comparatively, in the second scenario the *NSE* values during the verification period (from 2008 to
2012) are 0.81 and 0.82 for the two basins, respectively. The *NSE* values in the second scenario
are consistently higher than that in the first scenario in the two basins. For the UYLR in the second
scenario (Figure 6d), the *NSE* value during the verification period is significantly greater than that





in the first scenario. Figure 6(b) also shows that the HIMS hydrological model significantly
underestimates the flow peaks during the summer of 2010 and 2012 when calibrated by 5 years of
data from 2003 to 2007. The disagreement between the observed and simulated flow peaks is partly
because the magnitudes of flood events during the calibration period are all smaller than that during
the verification period and the HIMS hydrological model cannot be well trained during the
calibration period. Therefore, when using a short length precipitation data as input for a
hydrological model, the accuracy of streamflow simulation could be limited, especially when
precipitation data used for calibration cannot cover the flood and drought conditions of a basin.
However, when the HIMS hydrological model is calibrated by the longer dataset from 1983 to
1997 as it is shown in Figures 6c and d, there is a greater potential that the characteristics of
extreme events can be captured by the hydrological model than only using 5 years of data from
2003 to 2007. Given the availability of long-term precipitation records (over 33 years) provided
by PERSIANN-CDR product, the extreme events in the historical period could be well captured
by a hydrological model. Therefore, using such a product with long-term records, the confidence
of simulating streamflow over the Tibetan Plateau will correspondingly increase.






Figure 6. The simulated daily streamflow (red) forced by PERSIANN-CDR rainfall product in

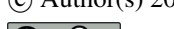



different scenarios and the observed daily streamflow (black) at the outlets of the upper Yangtze
River Basin and upper Yellow River Basin. (a) and (b) is the scenario that the period 2003 to 2007
is used for calibration and 2008 to 2012 for verification. (c) and (d) is the scenario that the period
1983 to 1997 is used for calibration and 2008 to 2012 for verification

**5.  Summary**

As it is compared to radar-based precipitation measurement and gauge networks, the main

advantage of satellite-based precipitation estimate is the broader coverage at global scale. This
allows a comprehensive understanding of the driving force of hydrologic cycle, especially for the
gauge sparse area. To verify the accuracy of satellite-based precipitation estimate products, the
comparison with ground observation is necessary. However, in gauge sparse area, a direct
comparison on precipitation temporal and spatial variation will be arguable due to the limited
gauge information. This study provides an alternative way to evaluate satellite-based precipitation
products by forcing both rainfall estimates from satellite and limited gauge network into
hydrological model. Given the confidence in streamflow measurements, which are more reliable
and well monitored than the limited ground-based rainfall measurements, the comparison of
simulated streamflow enables an indirect way to evaluate satellite-based precipitation products.

In this study, PERSIANN-CDR precipitation and gauge-based precipitation have good

agreements in the UYLR, while the two datasets have different values in the UYZR. Streamflow
simulation capabilities of PERSIANN-CDR precipitation and gauge-based precipitation are
evaluated as the inputs of the HIMS hydrologic model in the two basins. Both the two datasets
have similar good performances in the UYLR, while PERSIANN-CDR precipitation has slightly
better performance than gauge-based precipitation in the UYZR. Gauge-based precipitation tends





to produce larger streamflow in wet season in the UYZR. This indicates that in the UYZR, a sparse
gauge network are not a completely reliable reference for water resources management and
operation due to the fact that the locations of the limited gauge stations cannot be representative
for measuring the precipitation patterns at the river basin scale.

Lack of rainfall gauge stations has brought great challenge to hydrological and climate studies

over the Tibetan Plateau (e.g., Yao et al., 2012; Zhang et al., 2013). Based on the demonstration in
this study that PERSIANN-CDR is able to produce reasonably good streamflow in the UYZR and
UYLR as compared to observed streamflow, we can speculated that PERSIANN-CDR rainfall
product has the potential to be a useful dataset and alternative of the sparse gauge network for
future climate change and hydrological studies on the Tibetan Plateau considering the needs for
long-term (more than 33 years) and high resolution records.

**Data Availability**
The PERSIANN-CDR data used in this study is available from the NOAA CDR Program
(https://www.ncdc.noaa.gov/cdr/atmospheric/precipitation-persiann-cdr)    and    gauge    data    is
retrieved from China Meteorological Administration database (http://www.cma.gov.cn/en2014/
and http://data.cma.cn/). The data is public accessible by user registration from the providing
agencies.

**Acknowledgements**
This research was supported by the Natural Science Foundation of China (41571024, 41330529,
41201034), the program for "Bingwei" Excellent Talents in Institute of Geographic Sciences and
Natural Resources Research, CAS (Project No.2013RC202) and the NOAA NCDC/Climate Data





Record program (Prime Award NA09NES440006).

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

**Figure Captain**
Figure 1. The selected river basins (the upper Yellow River and Yangtze River Basin) on the
Tibetan Plateau and location of rainfall stations and river outlets.
Figure 2. The monthly average runoff observed at the river outlet of the upper Yangtze River and
Yellow River Basin, and the precipitation data retrieved from PERSIANN-CDR and ground-based
observation.
Figure 3. The calculated CDF of rainfall from PERSIANN-CDR and ground-based observation in
the upper Yangtze River Basin and upper Yellow River Basin.
Figure 4. The comparison between the simulated daily streamflow (red) with PERSIANN-CDR
and ground-based precipitation and the observed data (black) at the outlets of the upper Yangtze
River Basin (a and b) and upper Yellow River Basin (c and d).
Figure 5.The comparison between the observed streamflow (black) and the simulated streamflow
using PERSIANN-CDR (blue) and ground-based precipitation data (red) in the upper Yangtze
River Basin and upper Yellow River Basin.
Figure 6. The simulated daily streamflow (red) forced by PERSIANN-CDR rainfall product in
different scenarios and the observed daily streamflow (black) at the outlets of the upper Yangtze
River Basin and upper Yellow River Basin. (a) and (b) is the scenario that the period 2003 to 2007
is used for calibration and 2008 to 2012 for verification. (c) and (d) is the scenario that the period
1983 to 1997 is used for calibration and 2008 to 2012 for verification