# Peer review of "Hydrol. Earth Syst. Sci. Discuss., doi:10.5194/hess-2016-282, 2016 Manuscript under review for journal Hydrol. Earth Syst. Sci."

_Hydrology and Earth System Sciences, 2016_

## Referee Comment (RC1) · Anonymous Referee #1 · 26 Jul 2016

The hydrological modelling in the river basins over the Tibetan Plateau is always difficult, due to the limitation of available datasets, and the existence of cryosphere components (e.g., snow, glacier, and frozen soil). For the former, the authors made great efforts to simulate the streamflows over the upper Yellow and Yangtze river basins, with a newly developed daily satellite precipitation product by comparing to available gauge observations. For the two studied river basins, the upper Yangtze River Basin has very sparse precipitation observational stations, comparing to the upper Yellow River Basin. As expected, the performance of HIMS hydrological model forced by the PERSIANN-CDR satellite precipitation performs promising over the upper Yangtze

River Basin where the ground observations are very poor. In addition, the length (33 years) of PERSIANN-CDR makes it helpful in the calibration of hydrological model. In general, the manuscript is interesting to me and can be accepted after revisions.

Comments: (1) We can get limited knowledge if only one precipitation product is investigated. Considering the special length of precipitation datasets, suggest adding a similar one, the Global Land Data Assimilation System (GLDAS) precipitation for comparision. You may read (but not limited to) the following papers as a referebce. Gottschalck et al. (2005), J. Gottschalck, J. Meng, M. Rodell, P. Houser, Analysis of multiple precipitation products and preliminary assessment of their impact on global land data assimilation system land surface states, J. Hydrometeorol., 6 (2005), pp. 573–598 Wang et al. (2011), Evaluation and application of a fineresolution global data set in a semiarid mesoscale river basin with a distributed biosphere hydrological model, J. Geophys. Res., 116, D21108.

(2) Having better spatial distributions is a big merit of satellite-based precipitation product, comparing to the sparse ground-based observational sites over the Tibetan Plateau. Suggest adding the Figures of precipitation in their spatial distributions if possible.

(3) It is hard to compare the hydrological model's performance with only the basin-integrated streamflows. Suggest adding the comparisons of simulated evapotranspirations (ET) as well, to confirm the improvements of internal processes besides the final discharge outputs. For the ET estimation over the two river basins, suggest reading (but not limited to) the following papers: Zhang, Y. et al. (2007), Trends in pan evaporation and reference and actual evapotranspiration across the Tibetan Plateau, J. Geophys. Res., 112, D12110. Xue et al. (2013), Evaluation of evapotranspiration estimates for two river basins in Tibetan Plateau by a water balance method, Journal of Hydrology, 492, 290-297. Li et al. (2014), Seasonal evapotranspiration changes (1983–2006) of four large basins on the Tibetan Plateau, J. Geophys. Res. Atmos., 119, 13079–13095.

(4) Lack of frozen soil parametrization in HIMS may largely affect the simulated seasonal variation of water balance components (e.g., streamflow and evapotranspiration). It may bring certain uncertainties in the discharge comparisons by different precipitation inputs. To address the modelling issue may be out of the scope of this paper, but you can discuss the limitations/uncertainties in the "Summary" section.

(5) Line 233: please add the name of two basins here.

(6) Line 252, "have similar values": please specify the values here.

(7) Line 450: change "are" to "is"; replace "completely" with a more suitable word.

---

## Author Comment (AC1) · 4 Aug 2016

(1) We can get limited knowledge if only one precipitation product is investigated. Considering the special length of precipitation datasets, suggest adding a similar one, the Global Land Data Assimilation System (GLDAS) precipitation for comparision. You may read (but not limited to) the following papers as a reference. Gottschalck et al. (2005), J. Gottschalck, J. Meng, M. Rodell, P. Houser, Analysis of multiple precipitation products and preliminary assessment of their impact on global land data assimilation system land surface states, J. Hydrometeorol., 6 (2005), pp. 573–598 Wang et al.

(2011), Evaluation and application of a fine resolution global data set in a semiarid mesoscale river basin with a distributed biosphere hydrological model, J. Geophys. Res., 116, D21108.

Answer: Thank you for your suggestion. We totally agree with the comment. GLDAS precipitation also has a long length of data record. We will add the GLDAS precipitation as input to run the hydrologic model and make some comparison in the revised manuscript.

(2) Having better spatial distributions is a big merit of satellite-based precipitation product, comparing to the sparse ground-based observational sites over the Tibetan Plateau. Suggest adding the Figures of precipitation in their spatial distributions if possible.

Answer: Thank you for your suggestion. We agree with the comment. Adding some figures about the spatial distribution of PERSIANN-CDR precipitation is a good way to show the big merit of PERSIANN-CDR precipitation product. We will add the spatial distribution figures of PERSIANN-CDR precipitation in the revised manuscript.

(3) It is hard to compare the hydrological model's performance with only the basin integrated streamflows. Suggest adding the comparisons of simulated evapotranspirations (ET) as well, to confirm the improvements of internal processes besides the final discharge outputs. For the ET estimation over the two river basins, suggest reading (but not limited to) the following papers: Zhang, Y. et al. (2007), Trends in pan evaporation and reference and actual evapotranspiration across the Tibetan Plateau, J. Geophys. Res., 112, D12110. Xue et al. (2013), Evaluation of evapotranspiration estimates for two river basins in Tibetan Plateau by a water balance method, Journal of Hydrology, 492, 290-297. Li et al. (2014), Seasonal evapotranspiration changes (1983–2006) of four large basins on the Tibetan Plateau, J. Geophys. Res. Atmos., 119, 13079–13095.

Answer: Thank you for your suggestion. We agree with the comment. Adding ET comparisons can be a good supplement to verify hydrological model's performance.

HESSD
For hydrologic modeling, the improvements of internal processes besides the final discharge outputs are also important. We will add the ET comparison results in the revised manuscript.

(4) Lack of frozen soil parametrization in HIMS may largely affect the simulated seasonal variation of water balance components (e.g., streamflow and evapotranspiration). It may bring certain uncertainties in the discharge comparisons by different precipitation inputs. To address the modelling issue may be out of the scope of this paper, but you can discuss the limitations/uncertainties in the "Summary" section.

Answer: Thank you for your suggestion. Lack of frozen soil parametrization in HIMS definitely will affect the simulated seasonal variation of water balance components. Actually, we find that both PERSIANN-CDR and gauge-based precipitation generate smaller streamflow in dry season, which probably is due to the lack of proper algorithm in the HIMS model to handle frozen soil. We will add some discussion about the limitations of frozen soil simulation in the "Summary" section in the revised manuscript.

(5) Line 233: please add the name of two basins here. Answer: Thank you for your suggestion. We will add the basin name in the revised manuscript.

(6) Line 252, "have similar values": please specify the values here. Answer: Thank you for your suggestion. We will add the values in the revised manuscript.

(7) Line 450: change "are" to "is"; replace "completely" with a more suitable word. Answer: Thank you for your suggestion. We will improve the grammar in the revised manuscript.

HESSD
**Discussion** paper

---

## Short Comment (SC1) · 20 Sep 2016

Comments In the manuscript, entitled "Evaluating the streamflow simulation capability of PERSIANN-CDR daily rainfall products in two river basins on the Tibet Plateau", authors demonstrated an application study of a new satellite-based precipitation database and comparison with the precipitation from gauge-network. The study areas are on the Tibet Plateau and the gauge density is very sparse, which may not be a reliable data source for streamflow simulation and water resources management. The philosophy authors applies is to conduct evaluation via streamflow simulation from both precipitation sources and compare the simulations with streamflow gauge observation, which is believed to be more reliable than rain-gauges with regard to data length, accuracy, and continuity. The experiments are well designed and conducted, and the manuscript reads well. The following comments are suggested for author's consideration. The previous reviewer #1 made a couple suggestive comments and I agree with most of the comments by reviewer #1. In details, (i) a comparison can be added to further strengthen the comparison. (ii) the evaporation simulation can also serve as the same logic to support authors' arguments. After all, the streamflow and evaporation are two of the major components of water cycle. The hydrological model should be able to provide such information. (iii) In author's reply to reviewer #1, authors also agree to provide the evaporation simulation/comparison in the revised manuscript. I am also interested to see the simulation results and comparison with other data sources. Is there a diagram or figure to illustrate the flow chart/conceptual configuration of the used HIMS hydrological model? By only reading text, reviewer finds it not intuitive on the model configuration. In addition, the manuscript still has few minor/editing issues that should be fixed before publication. In details: 1. Line 208-209: should be "There are two stopping criteria used in the SCE-UA algorithm "2. Line 212-213: suggest to add population size. 3. Line 231: there is an extra period. 4. Line 236: should be "the runoff coefficients are 0.29 for both PERSIANN-CDR and Gauge..." 5. Line 251: missing comma after "Aug." 6. Line 254: missing "the" before "average annual amounts" 7. Line 281: should be "two data sources". Basically, two datasets are same type as precipitation measures. 8. Line 301: replace "two basin" with specific names since it is the first sentence of a paragraph. 9. Line 360: there is an extra period 10. Line 360: should be "the bias between simulated and observed streamflow". 11. Line 411: do authors mean "partially"? 12. Line 413: replace "the calibration period" by "calibration" 13. Line 416: replace "flood and drought conditions" by "extreme conditions, such as flood and drought" 14. Line 418: add parentheses to Figure subplot citations 15. Line 422: Last sentence maybe change to "Therefore, using such a product with long-term records as forcings to hydrological models, the confidence of simulated streamflow over the TB

---

## Referee Comment (RC2) · Anonymous Referee #2 · 8 Nov 2016

In this manuscript, authors presented an application of a precipitation estimate product based on satellite (PERSIANN-CDR) on gauge-sparse area, in which the accuracy of PERSIANN-CDR on two river basins on the Tibet Plataea of China are evaluated in terms of the simulated streamflow using a conceptual hydrological model. In the two river basins, gauge or radar information is limited in mountainous area due to their distribution, coverage, and beam angle. Therefore, satellite information will be good alternative than other sources of information. Before practical uses, verification is needed so that decision makers and local agency can have certain level of confidence which source of information is the most reliable. The contributions of this paper are

two in reviewer's opinion: (1) it evaluates a recent develop long-term global precipitation dataset against gauge, and GLADAS (in the revised version attached to AC2), and demonstrates the accuracy of streamflow simulation for the three sources of information. (2) the provides a way of utilizing streamflow to verify precipitation products, since streamflow is more reliable in mountainous area. The approach author took in this manuscript can be applied in other gauge-limited area for verification study.

As mentioned by anonymous referee 1 and short comment reviewer, the comparison with other source of precipitation data will be beneficial to improve the manuscript. After all, the sore comparison between satellite precipitation with limited gauge network via streamflow cannot fully support the conclusion of satellite information is better than limited gauge network for the two river basins on TP. Adding other source of information, such as GLADAS, could be considered as a more comprehensive study. In addition, the frozen soil issue is common in conceptual hydrological model, regardless whether the model is distributed, semi-distributed, lump. However, this does not undermine the approach that authors are trying to propose and the message authors want to delivery. If using land-surface models instead of hydrological models, that will be another study that is out of the scope of this study. Last, reviewer think the length of data is very crucial in simulating the streamflow. As authors did in discussion, different lengths of calibration data are used to study the sensitivity of data. It is suggested that authors also mention this in the context that besides the accuracy of data, the length is also important.

In general, I noted that this manuscript has already been revised from its original submission through the a several open discussion processes. The comments given by anonymous reviewer 1 and short comment reviewer in previous open discussion phases are suggestive and important. I agree with anonymous referee 1 and short comment (SC) reviewer that the original submission suffered from not addressing those key points, including the evaporation, comparison with GLADAS precipitation, frozen soil issue, and some minor language issues. By comparing the original submission and
the revised version attached to authors' reply to SC1, I think the authors did a good job in addressing previous comments: the comparison of GLADAS is added, the evaporation and frozen soil issues are discussed since they are key element in TP area, and the presentation (grammar) has been improved.

Therefore, I think the revised version is suitable for prompt publication. The following are only minor editing issues that can be fixed in proof-reading or revise phase. (Line numbers refer to the revised version attached to AC2).

Line 42: "potential to be a reliable"

Line 97: missing "the" before United State

Line 99: "show"

Line 108: add "the" before "limited" and "precipitation"

Line: 110: "capabilities"

Line: 120: "relatively"

Line 122: CMORPH "start"

Line 253: "Hydrometeorology"

Line 350: replace "both" with "all"

Line 423: insert "a" before "previous study"

Line 480: "using only"

Line 510: replace "both" with "all"

Line 512: "have" Line 515: could "not be" fully Line 527: product "has" Line 528: insert "an" before alternative Line 528: replace "for" with "in"

---

## Author Response (AR1)

This document contains point-to-point reply to Referees and track change version of the manuscript. (Answerers and Changes are marked in blue)

The following are our replies *Referee #1 (anonymous), Referee #2 (SC), and Referee #3 (anonymous)*:

***Referee 1 comment and reply:***

(1) We can get limited knowledge if only one precipitation product is investigated. Considering the special length of precipitation datasets, suggest adding a similar one, the Global Land Data Assimilation System (GLDAS) precipitation for comparision. You may read (but not limited to) the following papers as a reference. Gottschalck et al. (2005), J. Gottschalck, J. Meng, M. Rodell, P. Houser, Analysis of multiple precipitation products and preliminary assessment of their impact on global land data assimilation system land surface states, J. Hydrometeorol., 6 (2005), pp. 573–598 Wang et al. (2011), Evaluation and application of a fine resolution global data set in a semiarid mesoscale river basin with a distributed biosphere hydrological model, J. Geophys. Res., 116, D21108.

Answer: Thank you for the comments. Following your suggestion, we have added the GLDAS precipitation to compare with gauge observation and satellite product. In the revised manuscript, ground-based precipitation, GLDAS precipitation and PERSIANNCDR precipitation are used as the inputs of HIMS hydrologic model for streamflow simulation in the two river basins over TP. All the figures, tables and descriptions have been updated to the three precipitation datasets. Generally, GLDAS and PERSIANNCDR precipitation have a good consistency. Please see the revised manuscript for detail. See blue texts in the revised manuscript Introduction, Methodology and Reference Section.

(2) Having better spatial distributions is a big merit of satellite-based precipitation product, comparing to the sparse ground-based observational sites over the Tibetan Plateau. Suggest adding the Figures of precipitation in their spatial distributions if possible.

Answer: Thank you for your suggestion. We have added the spatial distribution of the GLDAS precipitation and PERSIANN-CDR precipitation in the revised manuscript. Please see the new Figure 3 and corresponding texts for details.

(3) It is hard to compare the hydrological model's performance with only the basin integrated streamflows. Suggest adding the comparisons of simulated evapotranspiration (ET) as well, to confirm the improvements of internal processes besides the final discharge outputs. For the ET estimation over the two river basins, suggest reading (but not limited to) the following papers: Zhang, Y. et al. (2007), Trends in pan evaporation and reference and actual evapotranspiration across the Tibetan Plateau, J. Geophys. Res., 112, D12110. Xue et al. (2013), Evaluation of evapotranspiration estimates for two river basins in Tibetan Plateau by a water balance method, Journal of Hydrology, 492, 290-297. Li et al. (2014), Seasonal evapotranspiration changes (1983–2006) of four large basins on the Tibetan Plateau, J. Geophys. Res. Atmos., 119, 13079–13095.

Answer: Thank you for your suggestion. We totally agree that adding evapotranspiration (ET) comparisons can be a good supplement to verify hydrological model's performance. The following figure shows the simulated ET from ground-based precipitation, GLDAS precipitation and PERSIANN-CDR precipitation by HIMS hydrological model and different ET products from Jung (2010), Zhang K. et al. (2010) and PenmanMentieth-Leuning (Leuning et al., 2008; Zhang Y. et al., 2016). We tried to compare and judge the different ET estimations, but we find that we maybe do not have a reliable reference for ET comparisons, because large-scale ET cannot be measured directly. Generally, large-scale ET estimated by water balance equation is a good reference. However, rainfall gauge information is limited in the TP as we mentioned in the manuscript, and we cannot use the limited ground-based precipitation to calculate basin reference ET based on water balance equation. Similar philosophy applies to other data-sources of precipitation. In other words, we can either use GLDAS precipitation nor PERSIANN-CDR precipitation to calculate basin reference ET based on water balance equation, because it would be unfair to compare these ET values with ET simulation from ground-based precipitation by HIMS hydrologic model. The purposes of this manuscript are to evaluate the streamflow simulation capability of PERSIANNCDR daily rainfall product. Therefore, we prefer to not present the ET results in the manuscript to avoid using any non-reliable ET estimation as reference to evaluate any precipitation products. Readers who are interested in the ET simulation can see the following figure, since all the discussion processes are permanently stored online of HESS Journal. Generally, the following figure shows that the simulated ET from the three precipitation datasets by HIMS model have better consistency in the upper Yellow River basin than in the upper Yangtze River basin. ET from Jung (2010) and PML (Leuning et al., 2008; Zhang Y. et al., 2016) are significantly smaller than ET simulated by the three precipitation based on HIMS model. Jung M, Reichstein M, Ciais P, et al. Recent decline in the global land evapotranspiration trend due to limited moisture supply. Nature, 2010, 467(7318): 951-954. Zhang K, Kimball J S, Nemani R R, et al. A continuous satelliteâˇRderived global record of ˇland surface evapotranspiration from 1983 to 2006. Water Resources Research, 2010, 46(9). Leuning R, Zhang Y Q, Rajaud A, et al. A simple surface conductance model to estimate regional evaporation using MODIS leaf area index and the PenmanâˇRMon- ˇteith equation[J]. Water Resources Research, 2008, 44(10). Zhang Y, Peña-Arancibia J L, McVicar T R, et al. Multi-decadal trends in global terrestrial evapotranspiration and its components. Scientific reports, 2016, 6.

(4) Lack of frozen soil parametrization in HIMS may largely affect the simulated seasonal variation of water balance components (e.g., streamflow and evapotranspiration). It may bring certain uncertainties in the discharge comparisons by different precipitation inputs. To address the modelling issue may be out of the scope of this paper, but you can discuss the limitations/uncertainties in the "Summary" section.

Answer: Thank you for your suggestion. We agree that lack of frozen soil parameterization in HIMS definitely will affect the simulated seasonal variation of water balance components. Actually, we find that all the three precipitation datasets generate smaller streamflow in dry season, which probably is due to the lack of proper algorithm in the HIMS model to handle frozen soil. We have added some discussions about the limitations of frozen soil simulation in the conclusion section in the revised manuscript. Please see line 516-521 of the revised manuscript for detail.

(5) Line 233: please add the name of two basins here.

Answer: Thank you for your suggestion. We have added the basin name in the revised manuscript. Please see line 260 of the revised manuscript.

(6) Line 252, "have similar values": please specify the values here.

Answer: Thank you for your suggestion. We have added the values in the revised manuscript.

Please see line 262-266 of the revised manuscript.

(7) Line 450: change "are" to "is"; replace "completely" with a more suitable word.

Answer: Thank you for your suggestion. We have improved the grammar in the revised manuscript.

***Referee 2 comments and reply:***

In the manuscript, entitled "Evaluating the streamflow simulation capability of PERSIANN-CDR

daily rainfall products in two river basins on the Tibet Plateau", authors demonstrated an application study of a new satellite-based precipitation database and comparison with the precipitation from gauge-network. The study areas are on the Tibet Plateau and the gauge density is very sparse, which may not be a reliable data source for streamflow simulation and water resources management. The philosophy authors applies is to evaluate the streamflow simulation from both precipitation sources and compare the simulations with streamflow gauge observation, which is believed to be more reliable than rain-gauges with regard to data length, accuracy, and continuity. The experiments are well designed and conducted, and the manuscript reads well. The following comments are suggested for author's consideration.

The previous reviewer #1 made a couple suggestive comments and I agree with most of the comments by reviewer #1. In details, (i) a comparison can be added to further strengthen the comparison. (ii) the evaporation simulation can also serve as the same logic to support authors'

arguments. After all, the streamflow and evaporation are two of the major components of water cycle. The hydrological model should be able to provide such information. (iii) In author's reply to reviewer #1, authors also agree to provide the evaporation simulation/comparison in the revised manuscript. I am also interested to see the simulation results and comparison with other data sources.

Answer: Thank you for your suggestions. Your comments are in-line with Reviewer #1, and please refer our reply to Reviewer #1 for details. With respect to your three comments, the detailed responses are lists as follow: As our answers to first referee's comment, GLDAS precipitation has been added to compare with gauge information and PERSIANN-CDR precipitation. In addition, the spatial distribution is added to let readers have a vivid impression on two precipitation datasets.

The following figure shows the simulated ET from ground-based precipitation, GLDAS precipitation and PERSIANN-CDR precipitation by HIMS hydrological model and different ET products from Jung (2010), Zhang K. et al. (2010) and Penman-Mentieth-Leuning (Leuning et al., 2008; Zhang Y. et al.,

2016). Readers who are interested in the ET simulation can see the following figure, since all the discussion processes are permanently stored online of HESS Journal. Generally, the following figure shows that the simulated ET from the three precipitation datasets by HIMS model have better consistency in the upper Yellow River basin than in the upper Yangtze River basin. ET from Jung (2010) and PML (Leuning et al., 2008; Zhang Y. et al., 2016) are significantly smaller than ET

simulated by the three precipitation based on HIMS model. More discussion about ET simulation
please refer our reply to comments of referee #1, and also the corresponding contents in the
revised manuscript. We sincerely thank the reviewer's suggestive comment. The revised
manuscript should be more satisfying.

[Figure]

**Fig. 1.** Authors' Reply to Comments Figure 1

References:

Jung M, Reichstein M, Ciais P, et al. Recent decline in the global land evapotranspiration trend
due to limited moisture supply. Nature, 2010, 467(7318): 951-954.
Zhang K, Kimball J S, Nemani R R, et al. A continuous satellite derived global record of land
surface evapotranspiration from 1983 to 2006. Water Resources Research, 2010, 46(9).
Leuning R, Zhang Y Q, Rajaud A, et al. A simple surface conductance model to estimate
regional evaporation using MODIS leaf area index and the Penmanâ˘ RMon- ˘ teith equation[J].
Water Resources Research, 2008, 44(10).
Zhang Y, Peña-Arancibia J L, McVicar T R, et al. Multi-decadal trends in global terrestrial
evapotranspiration and its components. Scientific reports, 2016, 6.

Is there a diagram or figure to illustrate the flow chart/conceptual configuration of the used
HIMS hydrological model? By only reading text, reviewer finds it not intuitive on the model
configuration.
Answer: Thank you for your suggestion. We have added the conceptual configuration of the
used HIMS hydrological model. Please see line 227. In addition, the manuscript still has minor and
few editing issues that should be fixed before publication.

In details:
1. Line 208-209: should be "There are two stopping criteria used in the SCE-UA algorithm "
Answer: Fixed

2. Line 212-213: suggest to add population size.

Answer: Added

3. Line 231: there is an extra period.

Answer: Deleted

4. Line 236: should be "the runoff coefficients are 0.29 for both PERSIANN-CDR and Gauge. . ."

Answer: Fixed

5. Line 251: missing comma after "Aug."

Answer: added

6. Line 254: missing "the" before "average annual amounts"

Answer: added

7. Line 281: should be "two data sources". Basically, two datasets are same type as precipitation measures.

Answer: Fixed

8. Line 301: replace "two basin" with specific names since it is the first sentence of a paragraph.

Answer: Fixed

9. Line 360: there is an extra period Answer: Deleted 10. Line 360: should be "the bias between simulated and observed streamflow". Answer: Fixed 11. Line 411: do authors mean

"partially"?

Answer: Yes and Fixed

12. Line 413: replace "the calibration period" by "calibration"

Answer: Fixed

13. Line 416: replace "flood and drought conditions" by "extreme conditions, such as flood and drought"

Answer: Fixed

14. Line 418: add parentheses to Figure subplot citations

Answer: Added

15. Line 422: Last sentence maybe change to "Therefore, using such a product with long-term records as forcings to hydrological models, the confidence of simulated streamflow over the TB

area will correspondingly increase."

Answer: Changed.

 ***Referee 3 comments and reply:***

In this manuscript, authors presented an application of a precipitation estimate product based on
satellite (PERSIANN-CDR) on gauge-sparse area, in which the accuracy of PERSIANN-CDR on two
river basins on the Tibet Plataea of China are evaluated in terms of the simulated streamflow using
a conceptual hydrological model. In the two river basins, gauge or radar information is limited in
mountainous area due to their distribution, coverage, and beam angle. Therefore, satellite
information will be good alternative than other sources of information. Before practical uses,
verification is needed so that decision makers and local agency can have certain level of confidence
which source of information is the most reliable. The contributions of this paper are two in
reviewer's opinion: (1) it evaluates a recent develop long-term global precipitation dataset against
gauge, and GLADAS (in the revised version attached to AC2), and demonstrates the accuracy of
streamflow simulation for the three sources of information. (2) the provides a way of utilizing
streamflow to verify precipitation products, since streamflow is more reliable in mountainous area.
The approach author took in this manuscript can be applied in other gauge-limited area for
verification study. As mentioned by anonymous referee 1 and short comment reviewer (referee 2),
the comparison with other source of precipitation data will be beneficial to improve the manuscript.
After all, the sore comparison between satellite precipitation with limited gauge network via
streamflow cannot fully support the conclusion of satellite information is better than limited gauge
network for the two river basins on TP. Adding other source of information, such as GLADAS, could
be considered as a more comprehensive study. In addition, the frozen soil issue is common in
conceptual hydrological model, regardless whether the model is distributed, semi-distributed,
lump. However, this does not undermine the approach that authors are trying to propose and the
message authors want to delivery. If using land-surface models instead of hydrological models, that
will be another study that is out of the scope of this study. Last, reviewer think the length of data
is very crucial in simulating the streamflow. As authors did in discussion, different lengths of
calibration data are used to study the sensitivity of data. It is suggested that authors also mention
this in the context that besides the accuracy of data, the length is also important. In general, I noted
that this manuscript has already been revised from its original submission through the a several
open discussion processes. The comments given by anonymous reviewer 1 and short comment
reviewer in previous open discussion phases are suggestive and important. I agree with anonymous
referee 1 and short comment (SC) reviewer that the original submission suffered from not
addressing those key points, including the evaporation, comparison with GLADAS precipitation,
frozen soil issue, and some minor language issues. By comparing the original submission and the
revised version attached to authors' reply to SC1, I think the authors did a good job in addressing
previous comments: the comparison of GLADAS is added, the evaporation and frozen soil issues
are discussed since they are key element in TP area, and the presentation (grammar) has been
improved. Therefore, I think the revised version is suitable for prompt publication.
Answerer: Thanks for your review comments and inputs. The editing issues you summarized are all
fixed. Please refer to revised manuscript and the track changes version in this reply. Thank you.

The following are only minor editing issues that can be fixed in proof-reading or revise phase. (Line
numbers refer to the revised version attached to AC2).

Line 42: "potential to be a reliable" Line 97: missing "the" before United State

Answerer: Fixed

Line 99: "show" Line 108: add "the" before "limited" and "precipitation"

Answerer: Fixed

Line: 110: "capabilities"

Answerer: Fixed

Line: 120: "relatively"

Answerer: Fixed

Line 122: CMORPH "start"

Answerer: Fixed

Line 253: "Hydrometeorology"

Answerer: Fixed

Line 350: replace "both" with "all"

Answerer: Fixed

Line 423: insert "a" before "previous study"

Answerer: Fixed

Line 480: "using only"

Answerer: Fixed

Line 510: replace "both" with "all"

Answerer: Fixed

Line 512: "have"

Answerer: Fixed

Line 515: could "not be" fully

Answerer: Fixed

Line 527: product "has"

Answerer: Fixed

Line 528: insert "an" before alternative

Answerer: Fixed

Line 528: replace "for" with "in"

Answerer: Fixed

*The following is track change version:*

**Evaluating the streamflow simulation capability of PERSIANN-CDR**

**daily rainfall products in two river basins on the Tibetan Plateau**

Xiaomang Liu[1,2], Tiantian Yang[2], Koulin Hsu[2], Changming Liu[1] and

Soroosh Sorooshian[2]

1 Key Laboratory of Water Cycle & Related Land Surface Process, Institute of

[revised manuscript text omitted]

2.5˚ precipitation products. The performance of PERSIANN-CDR rainfall product has been tested and reported in different regions (e.g., Ashouri et al. 2015; Miao et al., 2015;

Zhu et al., 2016). Ashouri et al. (2015) found that PERSIANN-CDR precipitation is performing reasonably well when compared with radar and ground-based observations in the 1986 Sydney flood event of Australia and the 2005 Hurricane Katrina of the

United States. Zhu et al. (2016) compared precipitation estimation from PERSIANN-

CDR, TRMM-3B42-V7 and CMORPH over the Xiang and Qu River Basins in China and demonstrated the accuracy of PERSIANN-CDR. Miao et al. (2015) show that

[revised manuscript text omitted]

1 station per grid of 1°×1° in the two basins, respectively. The precipitation data in

GLDAS comes from three different sources: the Climate Prediction Center Merged

Analysis of Precipitation, Global Data Assimilation System, and the European Centre for Medium-Range Weather Forecasts (Rodell et al., 2004). The precipitation data used in GLDAS is a combination of reanalysis and observations, which is believed to have the advantages of different data sources (Gottschalck et al., 2005). In this study, the 1.0-degree-resolution GLDAS precipitation dataset is re-sampled into 0.25˚×0.25˚ grids and used as the input of streamflow simulations (http://ldas.gsfc.nasa.gov/gldas/). 
[revised manuscript text omitted]
 spatial distribution of average annual values of 1.0-degree- resolution GLDAS precipitation and 0.25-degree-resolution PERSIANN-CDR

precipitation. The spatial patterns of the two dataset are generally consistent with each other. Figure 4 shows the comparison of CDFs for basin-averaged daily gauge-based precipitation, GLDAS precipitation and PERSIANN-CDR precipitation in the UYZR

and UYLR from 1983 to 2012. At a given probability, GLDAS precipitation generally has the smallest values, followed by PERSIANN-CDR precipitation and gauge-based precipitation in the UYZR. In the UYLR, the CDFs of PERSIANN-CDR precipitation,

GLDAS precipitation and gauge-based precipitation show overall better agreement than that in the UYZR. Table 2 shows the average amounts of gauge-based precipitation,

GLDAS precipitation and PERSIANN-CDR precipitation. In the UYZR, the average annual precipitation is 436.4 mm from gauge-based data, 365.1 mm from GLDAS

dataset and 374.3 mm from PERSIANN-CDR product. Gauge-based annual precipitation is 16.6% larger than PERSIANN-CDR annual precipitation. In the UYLR, average annual amounts of gauge-based precipitation, GLDAS precipitation and

PERSIANN-CDR precipitation are similar, which are 550.2, 547.9 and 556.6 mm, respectively (Table 2).

[Figure]

Figure 3    The spatial distribution of average annual values of 1.0-degree-resolution

GLDAS precipitation (a) and 0.25-degree-resolution PERSIANN-CDR precipitation (b).

[Figure]

Figure 4    The calculated CDF of daily gauge-based precipitation, GLDAS

precipitation and PERSIANN-CDR precipitation in the upper Yangtze River Basin and upper Yellow River Basin.

Table 2 Average monthly precipitation and runoff in the upper Yangtze and Yellow

River basins

| Period | Upper Yangtze River | | | | Upper Yellow River | | | |
|---|---|---|---|---|---|---|---|---|
| | Rain_ Gauge | Rain_ GLDAS | Rain_ CDR | Runoff_ OBS | Rain_ Gauge | Rain_ GLDAS | Rain_ CDR | Runoff_ OBS |
| Jan. | 3.3 | 4.0 | 1.4 | 1.3 | 4.4 | 5.3 | 3.2 | 3.7 |
| Feb. | 3.4 | 4.8 | 2.5 | 1.2 | 6.5 | 7.5 | 5.2 | 3.7 |
| Mar. | 5.0 | 8.1 | 7.5 | 1.5 | 12.9 | 16.2 | 13.1 | 4.8 |
| Apr. | 10.2 | 16.2 | 14.6 | 3.0 | 23.7 | 28.0 | 25.0 | 7.7 |
| May | 37.9 | 34.6 | 38.2 | 5.6 | 62.9 | 62.3 | 65.3 | 11.9 |
| Jun. | 90.4 | 66.3 | 72.0 | 12.9 | 107.6 | 96.2 | 104.6 | 20.4 |
| Jul. | 105.8 | 87.6 | 87.8 | 21.6 | 113.5 | 110.3 | 111.8 | 29.6 |
| Aug. | 88.6 | 69.0 | 74.5 | 20.6 | 92.0 | 93.3 | 94.0 | 23.3 |
| Sep. | 66.9 | 49.8 | 53.2 | 16.0 | 83.4 | 83.7 | 84.4 | 22.2 |
| Oct. | 20.2 | 18.0 | 20.5 | 9.1 | 35.3 | 36.0 | 41.4 | 19.4 |
| Nov. | 2.5 | 3.9 | 1.7 | 3.5 | 5.0 | 5.8 | 7.3 | 10.0 |
| Dec. | 2.3 | 2.0 | 0.5 | 1.6 | 3.0 | 3.3 | 1.5 | 5.0 |
| May to Oct. | 409.7 | 325.3 | 346.1 | 85.8 | 494.6 | 481.8 | 501.4 | 126.8 |
| Annual | 436.4 | 364.3 | 374.3 | 98.0 | 550.2 | 547.9 | 556.6 | 161.8 |
| Ratio | 93.9 | 89.3 | 92.5 | 87.6 | 89.9 | 87.9 | 90.1 | 78.4 |

*Note: Rain_Gauge, Rain_GLDAS and Rain_CDR indicate gauge-based precipitation*

*GLDAS precipitation and PERSIANN-CDR precipitation (mm), respectively.*

*Runoff_OBS indicates observed runoff (mm). Ratio means the percentage of precipitation and streamflow during May to November to annual values.*

**3.3 Streamflow Simulation in the two basins**

Due to the previous mentioned concern that sparse gauge-network and its interpolation cannot perfectly describe the spatial and temporal rainfall characteristics at river basin scale, the alternative is to evaluate streamflow simulated instead of treating sparse gauge-network as reference. In this section, the streamflow simulated by gauge-based precipitation, GLDAS precipitation and PERSIANN-CDR precipitation are derived from HIMS, and compared with observed streamflow at the outlet in the UYZR and UYLR. The HIMS model is separately calibrated by maximizing the *NSE* between observed streamflow and simulated streamflow driven by gauge-based precipitation, GLDAS precipitation and PERSIANN-CDR precipitation from 1983 to 1997. Table 3 shows the calibrated parameter values of the HIMS model for the two basins. Figure 5 shows daily observed streamflow and simulated streamflow driven by gauge-based precipitation, GLDAS precipitation and PERSIANN-CDR precipitation for the two basins from 1983 to 2012. In the UYZR (Figure 5 a, b and c), the *NSE* values are 0.63 0.78 and 0.77 in the calibration period driven by gauge-based precipitation, GLDAS precipitation and PERSIANN-CDR precipitation respectively, while they are 0.60, 0.71 and 0.73 in the verification period, respectively. In both calibration and verification period, the *NSE* values from GLDAS precipitation and PERSIANN-CDR precipitation are greater than that from gauge-based precipitation, which indicates that using GLDAS precipitation and PERSIANN-CDR precipitation as input to HIMS model is able to generate more accurate streamflow than using gauge-based precipitation in the UYZR. In the UYLR (Figure 5 d, e and f), the *NSE* values between daily observed streamflow and simulated streamflow are 0.82,

0.78 and 0.80 in the calibration period driven by gauge-based precipitation, GLDAS

precipitation and PERSIANN-CDR precipitation, respectively. In the verification period, the *NSE* values are 0.81, 0.77 and 0.78 for the three types of data, respectively.

The high *NSE* value in both calibration and verification periods suggest that gauge- based precipitation, GLDAS precipitation and PERSIANN-CDR precipitation have similar performances as the drivers of streamflow simulation in the UYLR.

[Figure]

Figure 5. The comparison between the simulated daily streamflow (red) with ground- based, GLDAS and PERSIANN-CDR precipitation and the observed data (black) at the outlets of the upper Yangtze River Basin (a, b and c) and upper Yellow River Basin (d, e and f).

Table 3 Calibrated parameter values in the HIMS model for the upper Yangtze and

Yellow River basins.

| Basin | input | $SMSC$ | $R$ | $r$ | $L_a$ | $R_c$ | $C$ | $K_b$ | $C_1$ | $C_2$ |
|-------|-------|--------|-----|-----|-------|-------|-----|-------|-------|-------|
| | Gauge_based | 302.5 | 1.47 | 0.78 | 0.74 | 0.05 | 0.67 | 0.15 | 0.18 | 0.81 |
| Yangtze | GLDAS | 339.2 | 1.72 | 0.87 | 0.82 | 0.07 | 0.58 | 0.18 | 0.17 | 0.81 |
| | PERSIANN-CDR | 343.8 | 1.71 | 0.89 | 0.87 | 0.07 | 0.56 | 0.18 | 0.17 | 0.82 |
| | Gauge_based | 334.8 | 2.08 | 0.77 | 1.00 | 0.03 | 0.44 | 0.14 | 0.14 | 0.86 |
| Yellow | GLDAS | 332.5 | 2.10 | 0.76 | 1.02 | 0.03 | 0.39 | 0.14 | 0.15 | 0.85 |
| | PERSIANN-CDR | 342.1 | 2.01 | 0.73 | 0.98 | 0.05 | 0.45 | 0.14 | 0.12 | 0.88 |

Figure 6 and Table 4 compare the simulated and observed average monthly streamflow for the two basins. In the UYZR, the relative bias between observed streamflow and simulated streamflow driven by gauge-based precipitation is 10.3% in wet season, which suggests a considerable overestimate of streamflow. Comparably, the relative bias between observed streamflow and simulated streamflow driven by

GLDAS precipitation and PERSIANN-CDR precipitation is -1.5% and 0.5% in wet season, respectively. As compared with the wet season streamflow simulation results with gauge-based precipitation, the simulated streamflows driven by GLDAS

precipitation and PERSIANN-CDR precipitation are closer to the observed streamflow.

In dry season, streamflow simulations driven by gauge-based precipitation, GLDAS

precipitation and PERSIANN-CDR precipitation all underestimate streamflow with relative bias of -22.1%, -20.1% and -28.0% in the UYZR, respectively. In the UYLR, all the three precipitation products slightly overestimate the streamflow in wet season with relative bias of 2.6%, 1.8% and 2.9%, respectively. Similar to the results in the

UYZR, streamflow simulations driven by gauge-based precipitation, GLDAS

precipitation and PERSIANN-CDR precipitation have similar good performances in wet season in the UYLR. However, all the three precipitation products tend to produce smaller streamflow in dry season with relative bias of -33.1%, -26.9% and -27.6%, respectively. One of the reasons that gauge-based precipitation, GLDAS precipitation and PERSIANN-CDR precipitation generate smaller streamflow in dry season is the lack of complex method or proper algorithm in the HIMS model to handle frozen soil.

In dry season, when the amounts of precipitation and streamflow are small, streamflow melted from frozen soil can account for a significant proportion of total streamflow. In other words, the frozen soil melt could significantly influence the streamflow simulation results. The relative high bias of observed streamflow and simulated streamflow from all the three precipitation products could be due to the lack of proper modeling component in the HIMS hydrologic model that quantifies the frozen soil melting effects in dry season. However, the bias between simulated and observed streamflow is much smaller in wet season, when precipitation and streamflow are relatively large and streamflow melted from frozen soil accounts for a limited proportion in total streamflow.

Table 4. The performances of streamflow simulations driven by gauge-based precipitation, GLDAS precipitation and PERSIANN-CDR precipitation in the two basins

| | Upper Yangtze River | | | | | | | Upper Yellow River | | | | | | |
|---|---|---|---|---|---|---|---|---|---|---|---|---|---|---|
| Period | Q_obs | Qs_ gauge | Qs_ GLDAS | Qs_ CDR | Rb_ gauge | Rb_ GLDAS | Rb _ CDR | Q_obs | Qs_ gauge | Qs_ GLDAS | Qs_ CDR | Rb _ gauge | Rb_ GLDAS | Rb _ CDR |
| Jan. | 68.1 | 48.4 | 40.4 | 32.8 | -28.9 | -40.7 | -51.8 | 168.9 | 65.7 | 71.4 | 68.0 | -61.1 | -57.7 | -59.8 |
| Feb. | 68.3 | 32.7 | 30.2 | 24.9 | -52.1 | -55.8 | -63.5 | 168.3 | 61.6 | 67.6 | 60.5 | -63.4 | -59.8 | -64.1 |
| Mar. | 76.9 | 70.2 | 75.3 | 72.4 | -8.7 | -2.1 | -5.8 | 219.7 | 110.5 | 145.1 | 138.0 | -49.7 | -34.0 | -37.2 |
| Apr. | 158.6 | 153.2 | 158.3 | 147.5 | -3.4 | -0.2 | -7.0 | 352.0 | 299.0 | 311.5 | 302.5 | -15.1 | -11.5 | -14.0 |
| May | 289.2 | 253.5 | 262.1 | 273.4 | -12.3 | -9.4 | -5.5 | 543.6 | 512.9 | 514.9 | 524.9 | -5.7 | -5.3 | -3.4 |
| Jun. | 683.9 | 750.5 | 679.1 | 698.4 | 9.7 | -0.7 | 2.1 | 928.5 | 968.6 | 921.3 | 946.6 | 4.3 | -0.8 | 1.9 |
| Jul. | 1108.9 | 1306.9 | 1102.5 | 1111.4 | 17.9 | -0.6 | 0.2 | 1350.1 | 1386.6 | 1420.2 | 1431.3 | 2.7 | 5.2 | 6.0 |

| | | | | | | | | | | | | | | |
|---|---|---|---|---|---|---|---|---|---|---|---|---|---|---|
| Aug. | 1059.7 | 1204.0 | 1042.8 | 1063.2 | 13.6 | -1.6 | 0.3 | 1061.1 | 1141.4 | 1102.7 | 1088.5 | 7.6 | 3.9 | 2.6 |
| Sep. | 850.7 | 977.4 | 897.2 | 918.9 | 14.9 | 5.5 | 8.0 | 1009.6 | 1059.7 | 1062.6 | 1075.7 | 5.0 | 5.2 | 6.5 |
| Oct. | 469.4 | 428.1 | 407.2 | 420.1 | -8.8 | -13.3 | -10.5 | 883.7 | 859.1 | 861.3 | 876.5 | -2.8 | -2.5 | -0.8 |
| Nov. | 187.6 | 169.0 | 182.3 | 161.1 | -9.9 | -2.8 | -14.1 | 457.3 | 429.1 | 437.8 | 456.6 | -6.2 | -4.3 | -0.2 |
| Dec. | 84.5 | 28.2 | 27.5 | 24.5 | -66.7 | -67.5 | -71.0 | 227.0 | 100.7 | 132.8 | 127.5 | -55.7 | -41.5 | -43.9 |
| May-Oct. | 743.4 | 819.6 | 731.9 | 746.9 | 10.3 | -1.5 | 0.5 | 962.7 | 987.7 | 980.5 | 990.4 | 2.6 | 1.8 | 2.9 |
| Nov.-Apr. | 107.2 | 83.6 | 85.6 | 77.2 | -22.1 | -20.1 | -28.0 | 265.6 | 177.6 | 194.2 | 192.3 | -33.1 | -26.9 | -27.6 |
| Annual | 427.9 | 454.6 | 408.7 | 414.8 | 6.2 | -4.5 | -3.1 | 617.0 | 586.0 | 587.8 | 594.6 | -5.0 | -4.7 | -3.6 |

*Note: Q_obs indicates observed runoff (m³/s). Qs_gauge, Qs_GLDAS and Qs_CDR*

*indicate streamflow simulations (m³/s) driven by the gauge-based precipitation,*

*GLDAS precipitation and PERSIANN-CDR precipitation, respectively. Rb_gauge,*

*Rb_GLDAS and Rb_CDR indicate relative bias between observed streamflow and*

*simulated streamflow driven by the gauge-based precipitation, GLDAS precipitation*

*and PERSIANN-CDR precipitation, respectively.*

[revised manuscript text omitted]
. In addition, gauge-based precipitation, GLDAS precipitation and PERSIANN-CDR precipitation all generate smaller streamflow in dry season probably because of lack of frozen soil algorithm in HIMS model. This may bring certain uncertainties in the discharge comparisons by different precipitation inputs (Xue et al., 2013b). Further studies should be conducted to improve the frozen soil simulation of HIMS model.

Lack of rainfall gauge stations has brought great challenge to hydrological and climate studies over the Tibetan Plateau (e.g., Yao et al., 2012; Zhang et al., 2013). Based on the demonstration in this study that PERSIANN-CDR is able to produce reasonably good streamflow in the UYZR and UYLR as compared to observed streamflow, we can speculate that PERSIANN-CDR rainfall product has the potential to be a useful dataset and an alternative for sparse gauge network in climate change and hydrological studies on the Tibetan Plateau considering the needs for long-term (more than 33 years) and high resolution records.

**Acknowledgements**

This research was supported by the Natural Science Foundation of China (41330529, 41571024, 41201034), the program for ''Bingwei'' Excellent Talents in Institute of Geographic Sciences and Natural Resources Research, CAS (Project No.2013RC202), the NOAA NCDC/Climate Data Record program (Prime Award NA09NES440006) and the DOE (Prime Award # DE-IA0000018).